# Genotypic Variation of Purple Rice in Response to Shading in Yield, Anthocyanin Content, and Gene Expression

**DOI:** 10.3390/plants12132582

**Published:** 2023-07-07

**Authors:** Nantapat Danpreedanan, Supapohn Yamuangmorn, Sansanee Jamjod, Chanakan Prom-u-thai, Tonapha Pusadee

**Affiliations:** 1Department of Plant and Soil Sciences, Faculty of Agriculture, Chiang Mai University, Chiang Mai 50200, Thailand; liuw_nantapat1@hotmail.com (N.D.); sansanee.j@cmu.ac.th (S.J.); chanakan.p@cmu.ac.th (C.P.-u.-t.); 2Lanna Rice Research Center, Chiang Mai University, Chiang Mai 50100, Thailand; teaw_848113@hotmail.com

**Keywords:** shading stress, purple rice, anthocyanin content, yield, *OsDFR*, low light intensity

## Abstract

Purple rice (*Oryza sativa* L.) contains anthocyanin, which acts as an antioxidant and functional food for humans. The levels of anthocyanin growth and production in rice are mainly controlled by the availability of light. However, shade can affect anthocyanin biosynthesis genes. Therefore, the objective of this study was to determine the yield and anthocyanin content among four purple rice varieties, which provide the difference in colors of purple and green leaves. This study also evaluated gene expression affected by shading treatment to understand the relation of grain anthocyanin and expression level. This research was conducted using a split plot design using four levels of shading (levels of shading from anthesis to maturity) with three replications, no shading, 30% shading, 50% shading, and 70% shading, as the main plots and purple rice varieties as subplots, KJ CMU-107, K2, K4, and KDK10, from anthesis to maturity. Shading significantly decreased yield and yield components, but increased grain anthocyanin content. Nonetheless, the response of yield and grain anthocyanin content to shading did not show a significant different between purple and green leaf varieties. In addition, the level of *OsDFR* gene expression was different depending on the shading level in four rice varieties. The *OsDFR* gene presented the highest expression at shading levels of 30% for K4 and 50% for KDK10, while the expression of the *OsDFR* gene was not detected in the purple rice varieties with green leaves (KJ CMU-107 and K2). The response of grain anthocyanin and gene expression of *OsDFR* to light treatment did not show significantly differences between the purple and green leaf varieties, suggesting that the appearance of anthocyanin in leaves might be not related to anthocyanin synthesis in the grain. Taken together, the results suggest that some purple rice varieties were more suitable for planting under low light intensity based on a lower level of grain yield loss, strong shade tolerance, and high anthocyanin content in leaf and grain pericarp. However, it is necessary to explore the effects of light intensity on genes and intermediates in the anthocyanin synthesis pathway for further study.

## 1. Introduction

Purple rice is commonly grown in the northern and northeastern regions of Thailand. Purple color appears in different parts of the purple rice plant, such as the pericarp, leaf sheath, leaf blade, and petals [1]. Today, increasingly more health-conscious consumers and researchers are turning to various rice varieties with black and purple seed coats that contain anthocyanins [2]. These substances are beneficial for health due to their antioxidant properties, which can improve blood flow in small vessels and reduce the risk of cancer and viral infection [3].

Light is essential for plant growth and development, but excess high-energy UV irradiance can cause damage to a cell. Anthocyanin accumulation in plants benefits plants by enhancing resistance to UV stress [4]. Light is a factor involved in plant growth since light is the source of energy used by plants in photosynthesis to produce sugar and starch. In addition, light plays an important role in various physiological processes within plants, such as protein synthesis, transpiration, and growth [5]. Ultimately, proper light intensity improves plant growth, including through the accumulation of antioxidants, which plants can synthesize to protect against the harmful effects of UV rays [6]. Therefore, the synthesis of anthocyanin, which is an antioxidant in the genetic system of plants, increases with increasing light intensity to a certain level, then decreases when light intensity is too great [7].

Shading with a black shading net reduces light intensity on plants and filters light [8]. Low light reduces photosynthesis and reduces plant yield [9,10]. This factor causes plants to adapt to survive and increases photosynthesis sources, stimulating the production of more anthocyanin in plants compared to plants grown under normal light conditions. In addition, anthocyanin is degraded if the light intensity is coupled with too high a temperature [11]. A previous report showed that anthocyanins in shoot and rice grain are sensitive to low light intensity, which suggests that anthocyanins could serve as a target compound for investigating low-light stress conditions [12]. Nevertheless, the anthocyanin biosynthesis gene related to the appearance of anthocyanin in various rice varieties has not been determined.

Anthocyanin pigment combines to form compounds with the ability to dissolve in the water found in plants. Such pigments play beneficial roles in visual activity, cancer, heart disease, and age-related neurodegenerative disorders [3]. Anthocyanins have protective effects during plant development through absorbing excess UV light, preventing lipid peroxidation, and suppressing the activity of ROS. Plants have evolved such that the biosynthetic pathways of anthocyanins can resist various abiotic stresses including UV irradiation, drought, high salinity, and low temperature [13]. The factors affecting the amounts of anthocyanins are either internal or external. The internal factors are related to plant genetics, which can be studied through genes that help regulate anthocyanin biosynthesis. The functions of genes that work together can be divided into two types: structural genes and regulatory genes [14]. External factors are factors in the natural environment or the occurrence of chemical reactions that affect the stability of anthocyanins, e.g., pH, light, temperature, and nutrients [15]. Both internal and external factors are directly related to increases and decreases in anthocyanin [16]. A previous study found that shade or low light intensity can affect anthocyanin levels. For example, although shade was found to reduce the production of upland rice, the anthocyanin content increased [11].

Usually, rice is a green plant; however, the biosynthesis of anthocyanin can impart purple, red, and black colors in leaf blades, leaf sheaths, stigmas, and pericarps [17]. Previous research and a growing body of experimental evidence suggest that anthocyanin provides plants with physiological benefits. However, the relevant mechanism has not been fully determined. The results of both chlorophyll content analysis and transmission electron microscopy revealed that anthocyanin has a negative impact on the photosynthetic machinery of purple-leaf compared to green-leaf wild-type plants [18]. A purple leaf color is an important morphological marker and valued as an important trait for the study of rice domestication and breeding [19]. In addition, previous research found that shade stress in plants leads to an increased amount of chlorophyll (chlorophyll a, chlorophyll b, and total chlorophyll) corresponding to the higher level of shade [11]. However, a previous study found that in purple rice varieties with purple leaves, the relative gene expression of *OsPL6* led to greater anthocyanin accumulation than that in purple rice varieties with green leaves [18]. Thus, we studied the differences in purple rice with both purple-leaf and green-leaf varieties under low-light conditions.

Anthocyanin biosynthesis is regulated by structural genes, yet the expression intensity of those genes is controlled by the interaction of regulatory genes, which are encoded by transcription factors (TFs) known as the MBW complex. Rice tissues are mainly controlled by three factors, C (Chromogen), A (activator), and P (Purple, distributor), where C and A are essentially color-producing genes, and P is a tissue-specific regulator of both C and A [20]. A combination called *OsDFR* results in a red grain color [21]. Two classes of genes are required for anthocyanin biosynthesis, the structural genes encoding the enzymes that directly participate in the formation of anthocyanins and other flavonoids and the regulatory genes that control the transcription of structural genes [22]. A previous study showed that enzyme activities in the various branch pathways are highly regulated (Figure 1). Genes code for the enzyme dihydroflavonol 4-reductase (DFR), which is involved in the process of changing dihydroflavonols to leucoanthocyanidins, which are colorless compounds, before being synthesized into anthocyanidin as the main structure of anthocyanins in the next step [23]. Anthocyanin biosynthesis was also found to be controlled in response to different developmental and environmental cues [22]. Therefore, the present study aimed to determine the yield and anthocyanin content among four purple rice varieties that provide the difference in color of purple and green leaves. This study also evaluated gene expression affected by shading treatment to understand the relation of grain anthocyanin to the expression level. The variety that responds well to light stress in terms of stability of grain yield and anthocyanin content would be useful for breeders in selecting plant varieties as parents in the assembly of new varieties.

## 2. Results

### 2.1. Yield and Yield Components

Grain yield, straw yield, grain weight, and filled grain amounts were significantly affected by the interactions between shading treatment and rice variety (Figure 2). The shading treatment strongly decreased grain yield by 17% to 81% in all varieties compared to no shading treatment. However, the level of decrease was dependent on rice variety. Shading at 30% resulted in 44% and 31% decreases in KJ CMU-107 and KDK10, respectively, and 19% and 17% decreases in K2 and K4. Similarly, shading treatment significantly decreased the straw yield. The straw yields of KJ CMU-107 and KDK10 grown under shading at 30% were reduced by 8% and 15%, respectively, compared to no shading treatment, whereas shading treatment showed no significant difference in K2 and K4. The result of the 100-grain weight of all varieties steadily reduced by 11% to 36% with an increase in shading level compared to the control treatment. A reduction in filled grain was found in all rice varieties and was obviously reduced when plants were treated with the 50% and 70% shading treatments.

Table 1 shows the number of tillers per plant, number of panicles per plant, number of spikelets per panicle, panicle length, and culm length in response to the shading treatment performed on the four varieties. Shading treatment had no effect on number of tillers per plant, number of panicles per plant, number of spikelets per panicle, or panicle length; however, the result was significantly different depending on rice variety. For instance, the highest number of tillers and panicles per plant, number of spikelets per panicle, and panicle length were observed in K2, while the lowest values were found in KDK10. By contrast, culm length differed according to shading treatment and rice variety. Shading at 50% resulted in a 3% reduction in culm length compared to the control treatment. KJ CMU-107 was the tallest followed by KDK10, K2, and K4, in that order.

### 2.2. Total Anthocyanin and Total Chlorophyll Content

The total anthocyanin contents in the leaves of two purple-leaf varieties at days 7, 14, and 21 after shading treatment present a significant interaction between shading treatment and rice variety (Figure 3a–c). Overall, an increase in shading level increased anthocyanin contents in the leaves, with an increase of 30% to 66% in K4 and 30% to 54% in KDK10 compared to the control plants. In addition, the grain anthocyanin contents of the four varieties were found to be significantly different based on the interaction between shading treatment and variety (Figure 3d). Shading at 30% yielded no significant increase in grain anthocyanin content in KJ CMU-107, K4, or KDK10 compared to the control shading but did show a 22% increase in K2. Across all varieties, shading at 50% resulted in the highest grain anthocyanin content, with increases of approximately 48%, 51%, 77%, and 51% in KJ CMU-107, KDK10, K2, and K4, respectively, compared to the control treatment. Nonetheless, grain anthocyanin content decreased when plants were grown under 70% shade, which was observed for all varieties, except KDK10. However, the color of pericarp when put under 30%, 50%, and 70% shade became darker compared to the plants grown with no shading. We also observed that the shapes of the seeds became smaller and abnormal when subjected to shading at 70% (Figure 4).

There was a significant interaction between shading treatment and rice variety in terms of total leaf chlorophyll content at days 7, 14, and 21 after shading treatment (Figure 5). At day 7, the total chlorophyll content steadily increased from 30% to 70% based on the shading level. However, the magnitude of the chlorophyll response was different between the green and purple-leaf varieties (Figure 5a). Total chlorophyll contents in the green leaves of KJ CMU-107 and K2 increased by approximately 13% to 32% compared to that for the plant with no shading treatment. Meanwhile, an increase of 26% to 60% was observed in the purple-leaf varieties, K4 and KDK10. Similarly, at day 14, the response of leaf chlorophyll content to the shading level continued to increase with an increase in shading level (Figure 5b). Shading treatment from 30% to 70% increased the chlorophyll content by 15% to 37% and 22% to 37% in the green-leaf varieties of KJ CMU-107 and K2, respectively, compared to the control plant. However, K4 and KDK10 presented strong increases in chlorophyll content of 37% to 77%. In addition, shading treatment had an effect on leaf chlorophyll content at day 21 for all varieties (Figure 5c). The chlorophyll contents of KJ CMU-107 and K2 increased by 41% to 79% in plants grown under 30% to 70% shade compared to the control plants and increased by 31% to 82% in K4 and KDK10, respectively.

### 2.3. Expression of OsDFR

The expression of *OsDFR* in the leaves at day 14 after shading was detected only in purple-leaf varieties K4 and KDK10, but no expression was detected in the green-leaf varieties of KJ CMU-107 and K2 (Figure 5). The expression level of *OsDFR* was significantly affected by the interaction between shading treatment and rice variety (Figure 6). The treatments of shading at 30% and 50% resulted in high expression levels of *OsDFR* in K4, which were 18% and 35% higher, respectively, than those in plants with no shading. However, a 70% shading treatment decreased the expression level by 57%. In contrast, the expression level of *OsDFR* of KDK10 grown under 30% shading treatment was not different compared to the level under no shading treatment, but the expression levels did significantly decrease by 78% and 82% when the plants were grown under shading treatments of 50% and 70%, respectively.

## 3. Discussion

This study shows that yield and yield components significantly decreased with increased levels of shading. A shading level of 70% produced the lowest yield as the plants received the lowest light intensity. Based on the results, grain filling was strongly affected by low light intensity compared to normal light conditions. Moreover, shading stress decreased the dry matter accumulation of rice plants, as well as the grain weight. Plant productivity was found to be mainly controlled by the photosynthesis rate since reduced light intensity generally reduced the amount of source [11,25]. A previous study reported that shading stress reduces the supply of photosynthetic products, thereby remarkably decreasing starch biosynthesis in grains and postponing caryopsis development [26]. When plants were disrupted by shading at the heading stage, a decrease in sink capacity was observed, leading to a strong reduction in grain-filling rate and grain yield [9,27]. Meanwhile, shading reduced the energy and nutrient supply, which affected grain-filling progress, pollen germination, and tube elongation [28]. Therefore, the grain-filling progress of rice, especially for spikelets at the bottom and middle positions, was significantly decreased by shading after the heading stage.

The magnitude of the decline in rice production due to shade depends on the level of tolerance and growth phases of each variety [29]. The results of this experiment suggest that shading treatment interfered with the source of the plant, which reduced grain filling, but the sink storage remained the same. As a result, the seeds became more withered because the leaves were unable to synthesize enough carbohydrates to produce sufficient amounts of source for all the seeds. This study found an 11–36% reduction in grain weight across all rice varieties in the shading treatment. Nonetheless, the KDK10 variety showed more stable 100-grain weight under the various shade levels compared to other varieties. A rice variety with stable yield parameters would be useful for selecting rice varieties to be planted under natural low-light conditions in order to achieve higher productivity in the future.

Shading treatment significantly increased anthocyanin levels in both the leaf and grain pericarp of purple rice. This result indicates that light could be the main factor in the anthocyanin synthesis of rice plants grown under shade conditions, which is in accordance with previous reports [10,12,30]. This study is the first observation on the responses of anthocyanin synthesis to low light intensity among various rice varieties. Interestingly, this research shows that different varieties yielded different anthocyanin accumulation results. For instance, the level of grain anthocyanin content in K2 and K4 varieties responded less strongly to reduced light intensity compared to other varieties. Meanwhile, the severe condition of low light intensity did not increase anthocyanin in the rice grains. Under the lowest light intensity treatment (70% of shading), the anthocyanin contents of all rice varieties tended to decrease, except in KDK10. However, the mechanism and function of anthocyanins in each variety remain unclear. Even though anthocyanins are not directly involved in plant growth, they can protect against plant damage from abiotic stress. A previous study reported that abiotic stresses inhibits plant growth and reduces crop productivity and that plants produce anthocyanins after ROS signaling via the transcription of anthocyanin biosynthesis genes, enabling increased anthocyanin to alleviate plants under stress conditions [13,31]. Nonetheless, the increased anthocyanin content in leaf and grain samples in the present study was not found to be related to the stability of grain yield. This result agrees with the response of anthocyanin to the shading of green- and purple-leaf varieties. The appearance of leaf anthocyanin in the purple-leaf variety could not maintain stable yield productivity. This result suggests that light is a necessary factor for rice production, especially during the flowering stage because carbohydrate accumulation in grains mainly depends on the photosynthesis rate. Although anthocyanin synthesis in leaf and pericarp was not found to be correlated in this experiment, this phenomenon should be studied in further research on anthocyanin transport from leaf to pericarp in purple rice. The gene expression of *OsDFR* was induced by low light intensity, which is similar to the response of grain anthocyanin. Under increased shading levels, the gene expression and grain anthocyanin of K4 tended to increase, but this result contrasts with that of KDK10. This result suggests that *OsDFR* gene expression could alert anthocyanin synthesis in the grain but might differ by rice variety. Other genes, such as *OsANS*, were reported to have an impact on grain anthocyanin [32]. However, a previous study found that *OsC1* and *OsRb* are tissue-specific regulators that do not influence anthocyanin biosynthesis in the pericarp [33].

Dihydroflavonol 4-reductase (DFR) uses NADPH as a cofactor to catalyze the reduction of dihydroflavonols to their respective colorless, unstable leucoanthocyanidins, which are common precursors for anthocyanin and proanthocyanidin biosynthesis [34]. The results show that the *OsDFR* gene was expressed only in purple leaves in K4 and KDK10 since these genes are important genes in anthocyanin biosynthesis that contain the enzyme generation code dihydroflavonol 4-reductase (DFR) in the process of changing dihydroflavonols to leucoanthocyanidins, which is a colorless compound, before being synthesized into anthocyanidin as the main structure of anthocyanins [35]. The genes that encode DFR and related proteins have been isolated from many plant species and have been well characterized in terms of their functions [36]. The gene expression at a shading level of 50% in K4 featured the highest gene expression of *OsDFR* due to the acceleration of anthocyanin biosynthesis in plants caused by high light shading stress. However, the gene expression of *OsDFR* was decreased under a shading level of 70%, which indicates that the plants were exposed to too little light for anthocyanin biosynthesis to occur. The *OsDFR* gene may not be involved in anthocyanin synthesis in K4 and KDK10, which should be further studied in other genes if the genes involved in anthocyanin content in K4 and KDK10 are to be investigated. The *OsDFR* gene has the ability to synthesize dihydroflavonol into leucoanthocyanidin. These compounds have no color of their own, but in acidic environments and at elevated temperatures, they are converted to the color of anthocyanidins [37]. A previous study found that *OsDFR* and other anthocyanin biosynthesis genes become purple in the apiculi and stigmas [36]. However, in the present research, we studied the *OsDFR* gene with shading treatment in the leaf and grain pericarp. No correlation was found in either rice tissue, so we suggest studying the other genes that result in a direct correlation of shading treatment to explore the total anthocyanin content and relative gene expression. The results of this experiment suggest that the *OsDFR* gene may not directly affect rice leaves’ transformation into a purple color caused by anthocyanin accumulation and that there may be another gene that interacts with the *OsDFR* gene. A previous study found that the accumulation of anthocyanin in rice leaves is caused by the interaction of *OsC1*, *OsRb*, and *OsDFR* genes, which are key genes for determining anthocyanin biosynthesis in rice leaves [38]. Experiments on the relative gene expression affected by shading treatment may be of greater interest when the ANS gene is studied in purple rice in the future in order to directly identify the genes affecting the purple color caused by anthocyanin accumulation in purple rice leaves during shading.

## 4. Materials and Methods

### 4.1. Expression of OsDFR

Purple rice seeds were obtained from the Division of Agronomy, Faculty of Agriculture, Chiang Mai University. This study used four purple pericarp rice varieties consisting of two purple colors in the shoot and grain varieties, K4 and KDK10, and two non-purple colors in the shoot varieties, K2 and Kum Jao Morchor 107 (KJ CMU-107). The experiment used a split-plot design for three replications. Four rice varieties were used as a subplot, and four shading treatments were used as the main plot, arranged as follows: no shading (control) and 30%, 50%, and 70% light reduction compared to normal light.

The study was conducted in a glasshouse from August to December 2021, at the Agronomy Division, Faculty of Agriculture, Chiang Mai University, Chiang Mai, Thailand. Seeds (paddy rice) were sown after soaking in water overnight. Two-week-old seedlings were transferred into pots (30 cm in diameter and 25 cm in height). The plants were grown with a single seedling per hill and five hills per pot. The shading treatment was set up with black polypropylene netting that reduced the light intensity to 30%, 50%, and 70% of full light from anthesis to the mature stage in planting pots 30 cm in diameter and 25 cm in height, with 5 plants per pot. The light intensity was measured daily using a light meter (AS one LM-332, Osaka, Japan) above the plant canopy. A fertilizer formula of 15-15-15 (N-P-K) was applied at a rate of 3 g pot^−1^, and fertilizer in the form of urea (46-0-0) was applied at a rate of 5 g pot^−1^ before planting.

### 4.2. Yield Measurement and Sample Preparation

At maturity, five plants in each pot were harvested, from which the yield components were determined. The paddy rice was threshed manually and dried until the moisture content reached 14% before being weighed and measured for grain yield. The collected data on yield components consisted of tiller number plant^−1^, panicle number plant^−1^, percentage of filled grain, 100-grain weight, number of spikelets panicle^−1^, and culm and panicle lengths. The straw samples were weighed and recorded as straw dry weight after being dried in a hot air oven at 75 °C for 72 h [12]. Two fully expanded leaves from the top position were sub-sampled at days 7, 14, and 21 after shading treatment to determine chlorophyll content and gene expression, and the remaining samples were freeze-dried in a freeze dryer (CHRIST, Beta 2–8 LSCbasic, Harz, Germany) for 24 h and mechanically ground in a hammer mill (Scientific Technical Supplies D–6072 Dreieich, West, Germany) to determine anthocyanin content. The paddy rice was de-husked with a laboratory husking machine (Model P-1 from Ngek Huat Co., Ltd., Bangkok, Thailand) to produce brown rice for the determination of anthocyanin content.

### 4.3. Determination of Total Anthocyanin Content

The total anthocyanin contents in leaves at days 7, 14, and 21 after shading treatment and grains were determined using the modified pH-differential method of [39]. About 2.5 g of freeze-dried sample was added into a 25 mL tube. Then, we pipetted 24 mL of acidified methanol (70% methanol and 30% of 1.5 mol L^−1^ HCl) into the tube and shook the tube for 60 min. Exactly 2 mL of supernatant was added to two buffer solutions. The potassium chloride buffer (0.025 mol^−1^, pH 1) was measured for absorbance at a wavelength of 520 nm, and a sodium acetate buffer (0.400 mol L^−1^, pH 4.5) was measured for absorbance at a wavelength of 700 nm using a UV–VIS spectrophotometer (Biochrom Libra S22, Cambridge, UK). The absorbance was calculated in the diluted sample (A) as A = (A520 − A700) − (A520′ − A700′), where A520 and A700 are the absorbance values at 520 and 700 nm under pH 1.0, respectively, while A520′ and A700′ are the absorbance values at 520 and 700 nm under pH 4.5, respectively. The total anthocyanin content was calculated as follows:Total anthocyanin content = (A × MW × DF × 1000)/ε × L
where MW is the molecular weight of cyanidin-3-glucoside (449.2 g mol^−1^); DF is the dilution factor; ε is 26,900 M absorbance, and L is the cell path length (1 cm). Total anthocyanin content was expressed as milligrams of cyanidin-3-glucoside per kg of dry weight (mg kg^−1^) [12].

### 4.4. Determination of Total Chlorophyll Content

The total chlorophyll content in leaves at days 7, 14, and 21 after shading was determined using the method of Lichtenthaler and Wellburn, 1983. About 0.2 g of fresh sample was extracted with methanol under cold conditions in the darkness. The extraction process was performed with four replicates. The supernatant was warmed to room temperature before measurement in a spectrophotometer at 665 and 652 nm to assess chlorophyll a and chlorophyll b, respectively. The chlorophyll a (Chl a), b (Chl b), and total chlorophyll (Total Chl) content (mg L^−1^) were calculated as follows:Chl a = (16.29 × A665) − (8.54 × A652)
Chl b = (30.66 × A652) − (13.58 × A665)
Total Chl = (22.12 × A652) − (2.71 × A665)
where A665 and A652 are the absorbance of Chl a and Chl b, respectively. Chlorophyll contents were expressed as milligrams per gram of fresh weight [12].

### 4.5. Gene Expression of OsDFR

#### 4.5.1. RNA Extraction

Total RNA was extracted from the leaves at day 14 after shading using a PureLinkTM RNA Mini Kit (Invitrogen, Thermo Fisher Scientific, MA, USA). The fresh tissues were ground in liquid nitrogen. The extracted RNA samples were verified for quantity and quality using a nanodrop spectrophotometer and 1.5% agarose gel electrophoresis. The genomic DNA was removed from the RNA preparations in the following conditions: 1 µg of total RNA, 2 µL of 10× reaction buffer, 1 µL of DNase1, and 9–15 µL of DEPC-treated water (total volume 20 µL); next, the reaction was incubated at 37 °C for 30 min, 1 µL of 50 mM EDTA was added, and the mixture was incubated at 65 °C for 10 min. The total RNA was diluted to a 100 ng/µL concentration and used for the qRT-PCR experiments.

#### 4.5.2. cDNA Synthesis

The cDNA was synthesized from 1 µg of total RNA using a RevertAid first-strand cDNA synthesis Kit (Thermo scientific). PCR was carried out using 1 µg of total RNA (DNase I-treated), 1 µL of Oligo (dT)18, 2 µL of 10 mM dNTP mix, 6 µL of 5× RT buffer, 1 µL of RiboLock Rnase Inhibitor (20 U/µL), 1 μL of RevertAid RT (200 U/µL), and 1–5 µL of DEPC-treated water (total volume 30 µL) and a RevertAid first strand cDNA synthesis Kit (Thermo Scientific). Next, we incubated the reaction at 42 °C for 60 min and terminated the reaction by incubating it at 70 °C for 5 min. Then, we stored the mixture at −20 °C. The cDNA was verified for quantity and quality using a nanodrop spectrophotometer and 1.5% agarose gel electrophoresis.

#### 4.5.3. Gene Expression via Semi-Quantitative RT-PCR Analysis

The gene expression levels of *OsDFR* were analyzed via semi-quantitative RT-PCR using gene-specific primers of *OsDFR* and *OsActin1* (housekeeping gene) (Table 2). The PCR was performed in triplicate to amplify cDNA templates with *OsDFR* and OsActin using the following reaction: 2 µL of 1:20 diluted cDNA from 1 µg total RNA, 14 µL of water (ddH2O), 4 µL of 5× MyTaq Reaction Buffer, 0.2 µL of forward and reverse primer, 0.1 µL of 5 unit MyTaqTM HS DNA Polymerase (Bioline, London, UK), and 0.6 µL of DMSO (total volume 20 µL). The PCR was performed by denaturing the solution at 95 °C for 2 min, followed by 35 cycles at 95 °C for 30 s, primer annealing at 53 °C for 30 s, extension at 72 °C for 30 s, and a final extension at 72 °C for 5 min. For the semi-quantitative RT-PCR assays, the total amount of cDNA in the samples was standardized after the amount of actin mRNA was evaluated with an *OsActin* primer pair.

#### 4.5.4. Statistical Analysis

Statistical analyses were carried out using analysis of variance (ANOVA) (Statistic version 8.0 for Windows) for a spit plot design, and shading treatment and rice variety were used as the main plot and sub plot, respectively. The least significant difference (LSD) at *p* < 0.05 was used to compare the means for significant differences among parameters. The significance of the correlation coefficients was analyzed using Pearson correlations at *p* < 0.05.

The gene expression levels were analyzed by relative intensity compared to the reference gene (Actin) using the ImageJ software version 1.50i (Wayne Rasband National Institutes of Health, MD, USA). The relative intensity of gene expression was subjected to statistical analysis using the Statistica 8 software (analytical software SX, version 8, Tallahassee, FL, USA).

## 5. Conclusions

This study demonstrates that shading at the reproductive stage corresponds to a significant decrease in the yield productivity of all rice varieties. However, some varieties presented stable yield parameters such as grain filling, suggesting that there are potential useful characteristics for selecting varieties to be grown under natural low-light conditions. Shading increased the biosynthesis of anthocyanin and chlorophyll contents in the leaves of all rice varieties. Similarly, an increase in anthocyanin was found in the grain pericarp when the level of shading was increased, whereas the response was different between rice varieties. In addition, the responses of grain yield and increasing anthocyanin content to shading were not different between green- and purple-leaf varieties, suggesting that light intensity plays an important role in rice productivity and anthocyanin synthesis, especially during the flowering stage. Expression of the *OsDFR* gene, which aids in the biosynthesis of total anthocyanin content, was found only in K4 and KDK-10 plants with purple leaves. Nonetheless, the *OsDFR* gene expression levels of K4 and KDK-10 varieties showed different responses to shading treatments. Meanwhile, a correlation between expression and grain anthocyanin was not found in this study. This result suggests that under low-light conditions, other genes may be related to anthocyanin biosynthesis in rice grains. These results provide useful data for future studies to understand the biosynthesis of anthocyanin in purple rice grown under biotic stress conditions to maintain grain anthocyanin content and grain yield. This research suggests some varieties of purple rice can adapt and grow under low-light conditions, which may be useful for commercial cultivation of purple rice using an ideal shading technique to increase antioxidants from purple rice grains.

## Figures and Tables

**Figure 1 plants-12-02582-f001:**
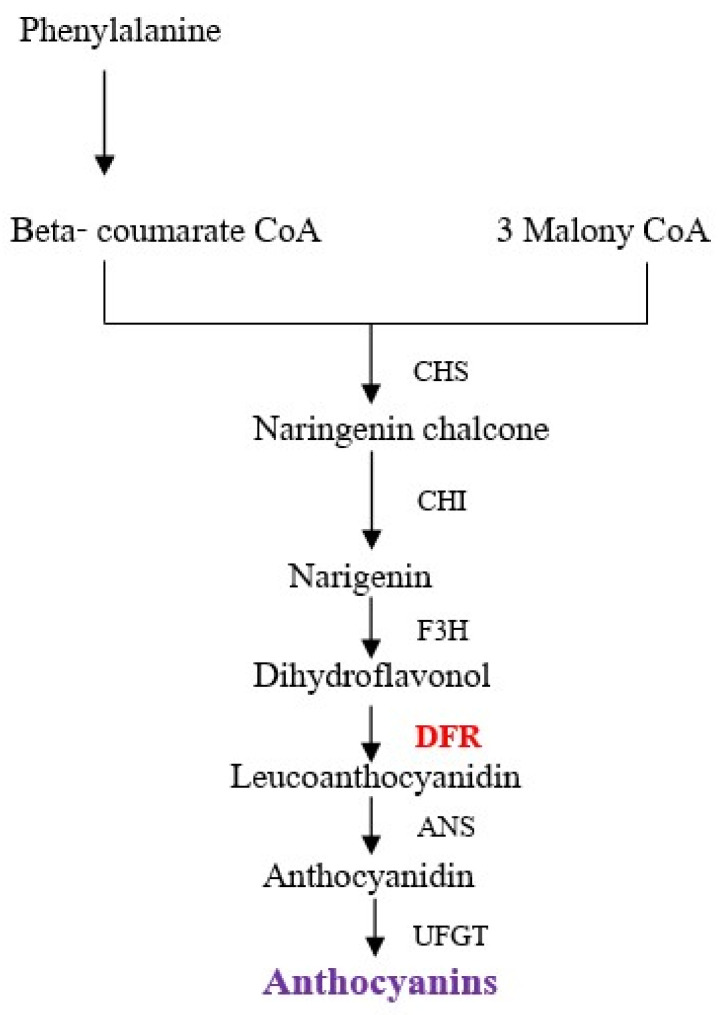
Pathway of anthocyanin biosynthesis. The *DRF* gene in red represents the candidate gene in the present study. Adapted from information in [24].

**Figure 2 plants-12-02582-f002:**
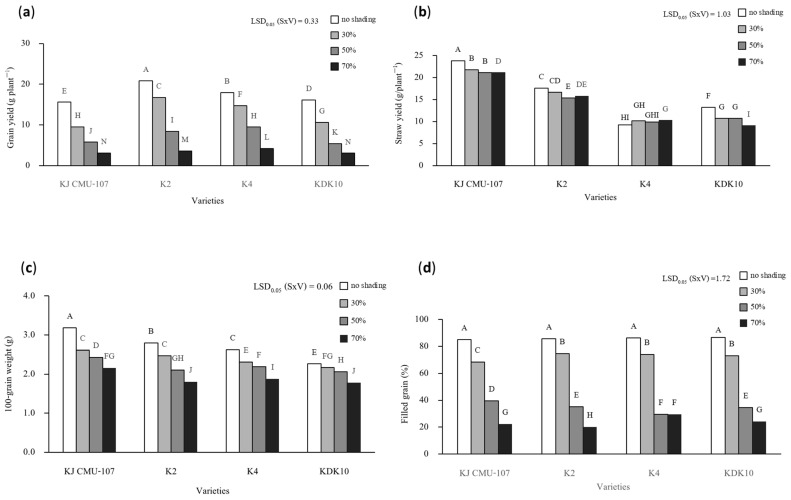
The interaction effects of yield and yield components between shading treatments (S); no shading with 30%, 50%, and 70% and purple rice varieties (V); KJ CMU-107, K2, K4, and KDK10. (**a**) Grain yield; (**b**) straw yield; (**c**) 100-grain weight; (**d**) filled grain. Different letters above bars indicate significant differences based on the least significant difference (LSD) at *p* < 0.05.

**Figure 3 plants-12-02582-f003:**
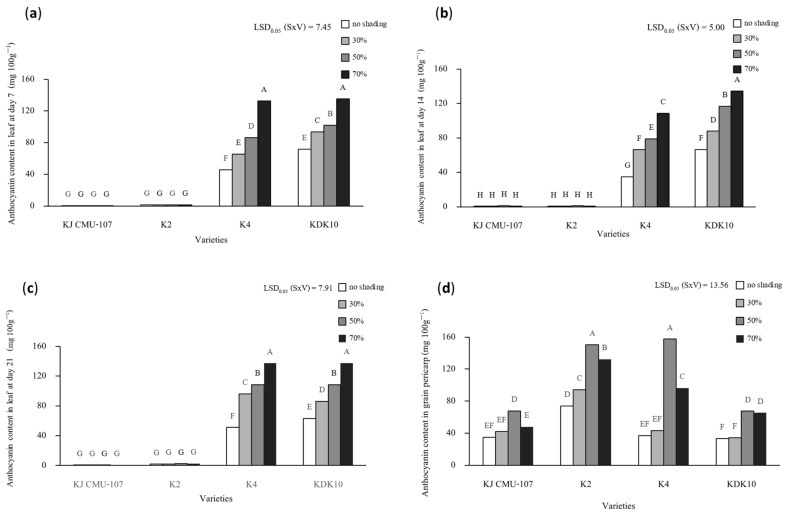
The interaction effect of total anthocyanin content between shading treatment (S); no shading treatment at 30%, 50%, and 70% with purple rice varieties (V); and KJ CMU-107, K2, K4, and KDK10. (**a**) Anthocyanin content in leaves at day 7; (**b**) anthocyanin content in leaves at day 14; (**c**) anthocyanin content in leaves at day 21; (**d**) anthocyanin content in grain pericarp. Different letters above the bars indicate significant differences based on the least significant difference (LSD) at *p* < 0.05.

**Figure 4 plants-12-02582-f004:**
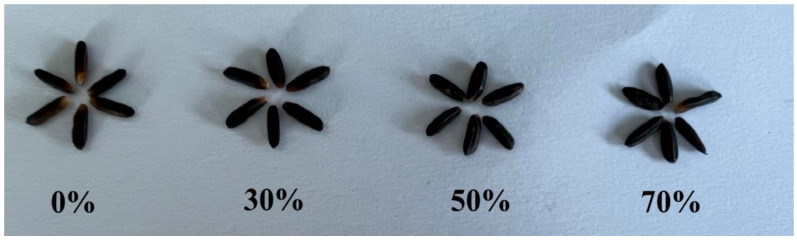
Grains of the K2 variety grown under varied shading: no shading and 30%, 50%, and 70% shading.

**Figure 5 plants-12-02582-f005:**
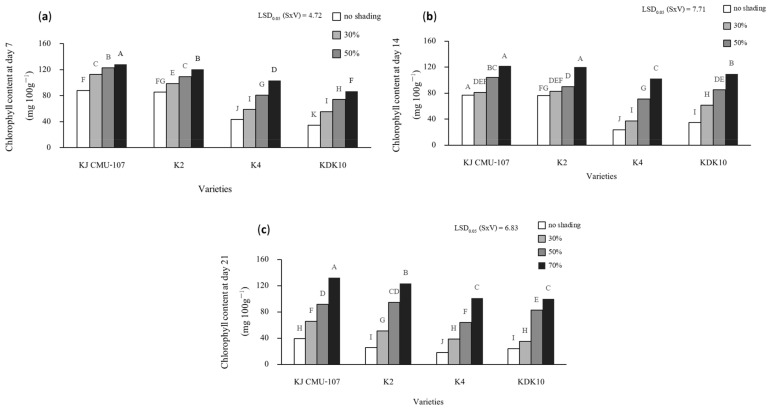
The interaction effect of total chlorophyll content between shading treatment (S); no shading at 30%, 50%, and 70 with purple rice varieties (V); and KJ CMU-107, K2, K4, and KDK10. (**a**) Chlorophyll content in leaves at day 7; (**b**) chlorophyll content in leaves at day 14; (**c**) chlorophyll content in leaves at day 21. Different letters above the bars indicate significant differences based on the least significant difference (LSD) at *p* < 0.05.

**Figure 6 plants-12-02582-f006:**
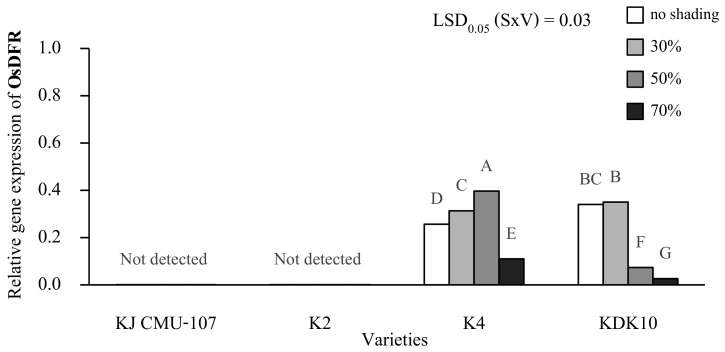
Relative gene expression of the *OsDFR* gene compared with *OsActin* (housekeeping gene) grown under varied shading; no shading, 30%, 50% and 70% with purple rice varieties; KJ CMU-107, K2, K4, and KDK10 in leaf at day 14 after shading. Different letters above the bars indicate significant differences based on the least significant difference (LSD) at *p* < 0.05.

**Table 1 plants-12-02582-t001:** Responses of number of tillers per plant, number of panicles per plant, number of spikelets per panicle, panicle length, and culm length.

	No. of Tillers Plant^−1^	No. of Panicles Plant^−1^	No. of Spikelets Panicle^−1^	Panicle Length (cm)	Culm Length (cm)
Shading treatment					
No shading	6.51 ± 0.27	5.96 ± 0.26	118.63 ± 0.24	24.65 ± 0.35	86.72 ± 1.66 AB
30%	6.42 ± 0.28	5.81 ± 0.28	118.93 ± 0.23	24.29 ± 0.34	87.10 ± 1.65 A
50%	6.28 ± 0.26	5.76 ± 0.29	119.06 ± 0.24	24.71 ± 0.31	84.25 ± 1.66 C
70%	6.25 ± 0.27	5.63 ± 0.28	119.24 ± 0.25	24.83 ± 0.32	85.40 ± 1.64 BC
Variety					
KJ CMU-107	6.53 ± 0.21 B	6.25 ± 0.22 B	98.57 ± 0.23 D	24.14 ± 0.33 B	121.98 ± 1.81 A
K2	8.23 ± 0.19 A	7.58 ± 0.23 A	140.97 ± 0.25 A	25.65 ± 0.34 A	71.07 ± 1.85 D
K4	4.93 ± 0.21 C	4.33 ± 0.22 D	122.08 ± 0.23 B	25.23 ± 0.33 A	74.18 ± 1.82 C
KDK10	6.43 ± 0.22 B	5.72 ± 0.21 C	113.97 ± 0.24 C	23.39 ± 0.32 C	76.23 ± 1.81 B
F-test					
Shading treatment (S)	ns	ns	ns	ns	*
LSD_0.05_ (S)					1.65
Variety (V)	***	***	***	***	***
LSD_0.05_ (V)	0.42	0.45	0.48	0.63	1.81
SxV	ns	ns	ns	ns	ns

The data were analyzed using F-tests (*: *p* < 0.05, ***: *p* < 0.001, ns: not significant *p* < 0.05). Different letters indicate the least significant differences in each parameter within the column at *p* < 0.05. The values are expressed as the mean ± SE.

**Table 2 plants-12-02582-t002:** Primer used for studying the gene expression of *OsDFR* and *OsActin* genes.

Gene	Primer Names and Sequences (5′➜3′)	References
*OsDFR*	*OsDFRF*: CGGGTTCAGGTTCAGGTACA	[40]
	*OsDFRR*: TGAAACCGGAGGGAGTAAC	[40]
*OsActin*	*OsActinF*: GACTCTGGTGATGGTGTCAGC	[41]
	*OsActinR*: GGCTGGAAGAGGACCTCAGG	[41]

F: forward primer; R: reverse primer.

## Data Availability

All data, tables, and figures in this manuscript are original.

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
