# Peer review of "Genotypic Variation of Purple Rice in Response to Shading in Yield, Anthocyanin Content, and Gene Expression"

_plants, 2023, doi:10.3390/plants12132582_

Round 1

Reviewer 1 Report

General Observations:

The authors conducted research on the purple rice content of anthocyanin acting as an antioxidant and its function as food for humans– on the premise that the levels of these antioxidants are controlled by the available light intensity and affects the biosynthesis genes of  anthocyanin. They used a split plot design with varying degree of shading parameters to assess the influence of shading-stress related to anthocyanin content, yield, and expression of the OsDFR gene on purple rice. The research is of great importance to the scientific community as this contributes to plant physiology and agronomic efforts of improving quality traits. However, the objectives were quite too obvious from the onset, as we know light and temperature has great effects on the yield and yield components–as well as quality traits. Kindly convince me some more apart from points already enumerated and thoughts from past studies, what is the uniqueness of this research with regards to yield and rice growth/development and quality of antioxidants on purple rice. We already know that the colour of the purple rice kennel is a morphological marker for the presence of antioxidant (anthocyanin), so ????. also is it feasible to commercialize purple rice cultivation using the ideal shading technique found in this research (as depicted in your conclusion–suggesting that some purple rice varieties were more suitable for planting under low light intensity based on a lower level of grain yield loss etc)? 

Overall the research is well written and I congratulate the authors for their effort. 

Minor corrections are required with concise sentences recommmed to avoid ambiguity.

Author Response

Reviewer 1

General Observations:

The authors conducted research on the purple rice content of anthocyanin acting as an antioxidant and its function as food for humans– on the premise that the levels of these antioxidants are controlled by the available light intensity and affects the biosynthesis genes of  anthocyanin. They used a split plot design with varying degree of shading parameters to assess the influence of shading-stress related to anthocyanin content, yield, and expression of the OsDFR gene on purple rice.

The research is of great importance to the scientific community as this contributes to plant physiology and agronomic efforts of improving quality traits.

However, the objectives were quite too obvious from the onset, as we know light and temperature has great effects on the yield and yield components–as well as quality traits. Kindly convince me some more apart from points already enumerated and thoughts from past studies, what is the uniqueness of this research with regards to yield and rice growth/development and quality of antioxidants on purple rice

Ans. Revised: Page 1 line 12-16

The objective of this study was to determine the yield and anthocyanin content among four purple rice varieties, which provide the different in colors of purple and green leaves. This study also evaluated the gene expression affecting by shading treatment to understand the relation of grain anthocyanin and the expression level.

Ans. Revised: Page 3 line 111-115

Therefore, the present study aimed to determine the yield and anthocyanin content among four purple rice varieties, which provide the different in colors of purple and green leaves. This study also evaluated the gene expression affecting by shading treatment to understand the relation of grain anthocyanin and the expression level.

also is it feasible to commercialize purple rice cultivation using the ideal shading technique found in this research (as depicted in your conclusion–suggesting that some purple rice varieties were more suitable for planting under low light intensity based on a lower level of grain yield loss etc)? 

Ans. Revised: Page 12 line 453-456

This research suggests some varieties of purple rice can adapt and grow under low light conditions, which may be useful for commercial cultivation of purple rice using an ideal shading technique to increase antioxidants from purple rice grains.

Reviewer 2 Report

In general, the quality of the article can be assessed as good, but there are some suggestions for its improvement
Hypothesis of the research should be specified
Line 327 - specify the formulation of the fertilizers
Line 417 - Type of ANOVA used for each statistical analysis
should be specified
Line 146-148 -described methods used for data analysis F-test, CV, these methods are not described in Materials and methods section
In my opinion, the term food is not used correctly in the article (for example lines 223, 52), I recommend replacing it with a more accurate term

Author Response

Reviewer 2

In general, the quality of the article can be assessed as good, but there are some suggestions for its improvement

  1. Line 327 - specify the formulation of the fertilizers

Ans. Revised: Page 10 line 332-334

A fertilizer formula of 15-15-15 (N-P-K) was applied at a rate of 3 g pot−1, and fertilizer in the form of urea (46-0-0) was applied at a rate of 5 g pot−1 before planting.

  1. Line 417 - Type of ANOVA used for each statistical analysis should be specified

Ans. Revised: Page 12 line 422-424

Statistical analyses were carried out using analysis of variance (ANOVA) (Statistic version 8.0 for Windows) for a spit plot design, shading treatment and rice variety were used as main plot and sub plot, respectively.

  1. Line 146-148 -described methods used for data analysis F-test, CV, these methods are not described in Materials and methods section

Ans. Revised:

Page 5 line 152 - 154

The data were analyzed using F-tests (*P < 0.05, ***P < 0.001, ns: not significant P < 0.05). Different letters indicate the least significant differences in each parameter within the column at P < 0.05. The values are expressed as the mean ± SE.

Page 12 line 423-425

Statistical analyses were carried out using analysis of variance (ANOVA) (Statistic version 8.0 for Windows) for a spit plot design, shading treatment and rice variety were used as main plot and sub plot, respectively.

  1. In my opinion, the term food is not used correctly in the article (for example lines 223, 52), I recommend replacing it with a more accurate term

Ans. Revised: Page 8 line – 227-229

Plant productivity was found to be mainly controlled by the photosynthesis rate since reduced light intensity generally reduced the amount of source.

Ans. Revised: Page 2 line – 55-56

This factor causes plants to adapt to survive and increase more sources.                          

Ans. Revised: Page 8 line – 242-243

As a result, the seeds became more withered because the leaves were unable to synthesize enough carbohydrates to produce sufficient amounts of source for all the seeds.   

Reviewer 3 Report

This manuscript contains useful and interesting results related to  purple rice responses to shading and should be acceptable after revision.  My comments are:

1.     Lines 15-17 change to “levels of shading from anthesis to maturity”

2.     Line 34 delete “very”

3.     Line 35 “improve blood vessels”, need to be more specific here.

4.     Starting at end of line 45:  Change to “increases with increasing light intensity to a certain level, then decreases when light intensity is too great.”

5.     Starting on line 48: Change to “Shading with black shading net reduces light intensity on plants and filters light [42].  Low light reduces photosynthesis and reduces plant yield [46, 62].  This factor causes. . .”

6.     Starting on line 61, change to “Anthocyanin pigment combines to form . . .Such pigments play beneficial .,. .”

7.     Starting at the end of line 67, change to “Factors affecting amount of anthocyanins are either internal or external.”

8.     Starting on line 79 change to “black colors in leaf blades, leaf sheaths, stigmas, and pericarps [28].  Previous research suggests that . . .”

9.     Line 83, change “explicated” to “determined”.

10.  Lines 176 and 178:  Round values to nearest whole number for the percentages.

11.  Line 236:  Change “light” to “carbohydrates”

12.  Line 452:  I am confused by this.  You say that “No new data were created or analyzed in this study”.  Was this data published elsewhere already?

Author Response

Reviewer 3

This manuscript contains useful and interesting results related to  purple rice responses to shading and should be acceptable after revision.  My comments are:

  1. Lines 15-17 change to “levels of shading from anthesis to maturity”

Ans. Revised: Page 1 line – 15-17

  1. Line 34 delete “very”

Ans. Revised: Page 1 line  40

These substances are very beneficial for health due to their antioxidant properties

  1. Line 35 “improve blood vessels”, need to be more specific here.                                   Ans. Revised: Page 1 line 41

improve the blood flow in the small vessels

  1. Starting at end of line 45:  Change to “increases with increasing light intensity to a certain level, then decreases when light intensity is too great.”

Ans. Revised: Page 2 line 51-52

  1. Starting on line 48: Change to “Shading with black shading net reduces light intensity on plants and filters light [42].  Low light reduces photosynthesis and reduces plant yield [46, 62].  This factor causes. . .”

Ans. Revised: Page 2 line 54-55

  1. Starting on line 61, change to “Anthocyanin pigment combines to form . . .Such pigments play beneficial .,. .”

Ans. Revised: Page 2 line  64-66

Anthocyanin pigment combines to form compounds with the ability to dissolve in the water found in plants. Such pigments play beneficial roles in visual activity, cancer, heart disease, and age-related neurodegenerative disorders.

  1. Starting at the end of line 67, change to “Factors affecting amount of anthocyanins are either internal or external.”

Ans. Revised: Page 2 line – 70-71

  1. Starting on line 79 change to “black colors in leaf blades, leaf sheaths, stigmas, and pericarps [28].  Previous research suggests that . . .”

Ans. Revised: Page 2 line – 80-82

  1. Line 83, change “explicated” to “determined”.

Ans. Revised: Page 2 line – 84

  1. Lines 176 and 178:  Round values to nearest whole number for the percentages.

Ans. Revised: Page 6 line – 181-184                                                                              

Shading treatment from 30% to 70% increased the chlorophyll content by 15% to 37% and 22% to 37% in the green-leaf varieties of KJ CMU-107 and K2, respectively, compared to the control plant. However, K4 and KDK10 presented a strong increase in chlorophyll content of 37% to 77%

  1. Line 236:  Change “light” to “carbohydrates”

Ans. Revised: Page 8 line – 241

  1. Line 452:  I am confused by this.  You say that “No new data were created or analyzed in this study”.  Was this data published elsewhere already?

Ans. Revised: Page 13 line – 462

T.P. is responsible for data keeping, and data are available upon request.

Round 2

Reviewer 1 Report

General Observations:

The authors conducted research on the purple rice content of anthocyanin acting as an antioxidant and its function as food for humans– on the premise that the levels of these antioxidants are controlled by the available light intensity and affects the biosynthesis genes of  anthocyanin. They used a split plot design with varying degree of shading parameters to assess the influence of shading-stress related to anthocyanin content, yield, and expression of the OsDFR gene on purple rice. The research is of great importance to the scientific community as this contributes to plant physiology and agronomic efforts of improving quality traits. The manuscript has largely been improved. However, data must be made available in a repository. it is not acceptable to only state that, data can be requested from an author.  Kindly reconsidered putting your data in the many repositories available, Zenodo, Mendeley Data, harvard Dataverse etc.

Congratulations for the efforts. 

The quality of english has largely improved, but some minor editting is reqiured. 

Author Response

The research is of great importance to the scientific community as this contributes to plant physiology and agronomic efforts of improving quality traits. The manuscript has largely been improved. However, data must be made available in a repository. it is not acceptable to only state that, data can be requested from an author.  Kindly reconsidered putting your data in the many repositories available, Zenodo, Mendeley Data, harvard Dataverse etc.

Ans: Data Availability Statement: All data, tables and figures in this manuscript are original.
